# The Global Association between Egg Intake and the Incidence and Mortality of Ischemic Heart Disease—An Ecological Study

**DOI:** 10.3390/ijerph20054138

**Published:** 2023-02-25

**Authors:** Norie Sugihara, Yoshiro Shirai, Tomoko Imai, Ayako Sezaki, Chisato Abe, Fumiya Kawase, Keiko Miyamoto, Ayaka Inden, Takumi Kato, Masayo Sanada, Hiroshi Shimokata

**Affiliations:** 1Faculty of Health and Social Services, Kanagawa University of Human Services, Yokosuka 238-8550, Japan; 2Faculty of Core Research, Ochanomizu University, Tokyo 112-8610, Japan; 3Department of Human Life and Environment, Kinjo Gakuin University, Nagoya 463-8521, Japan; 4Institute of Health and Nutrition, Nagoya University of Arts and Sciences, Nisshin 470-0196, Japan; 5Department of Food Science and Nutrition, Doshisha Women’s College of Liberal Arts, Kyoto 602-0893, Japan; 6Department of Food Science and Human Nutrition, Ryukoku University, Otsu 520-2194, Japan; 7Graduate School of Nutritional Sciences, Nagoya University of Arts and Sciences, Nisshin 470-0196, Japan; 8Department of Food and Nutrition, Tsu City College, Tsu 514-0112, Japan; 9Department of Nutrition, Asuke Hospital Aichi Prefectural Welfare Federation of Agricultural Cooperatives, Toyota 444-2351, Japan; 10Department of Nursing, Nagoya University of Arts and Sciences, Nagoya 460-0001, Japan; 11Clinical Nutrition Unit, Hamamatsu University Hospital, Hamamatsu 431-3192, Japan; 12Nutrition Division, Japanese Red Cross Aichi Medical Center Nagoya Daini Hospital, Nagoya 466-8650, Japan; 13Department of Nursing, Heisei College of Health Sciences, Gifu 501-1131, Japan

**Keywords:** egg intake, ischemic heart disease, ecological study, longitudinal analysis

## Abstract

The relationship between egg consumption and ischemic heart disease (IHD) remains controversial as there is still no clear answer regarding the relationship, with research limited to a few geographical regions. In the current study, we conducted a longitudinal analysis of the association between egg intake and IHD incidence (IHDi) and mortality (IHDd) using 28 years of international data from 1990 to 2018. Egg intake (g/day/capita) by country was obtained from the Global Dietary Database. Age-standard IHDi and IHDd rates per 100,000 subjects in each country were obtained from the 2019 Global Burden of Disease database. The analysis included a total of 142 countries with populations of at least one million, for which all data were available from 1990 to 2018. Eggs are consumed worldwide, and regional differences in consumption are also shown. Utilizing IHDi and IHDd as objective variables and egg intake as an explanatory variable, the analysis was conducted using linear mixed models, which controlled for inter- and intra-country variation from year to year. The results showed a significant negative association between egg intake, and IHDi (−0.253 ± 0.117, *p* < 0.05) and IHDd (−0.359 ± 0.137, *p* < 0.05). The analysis was carried out using R 4.0.5. The results suggest that adequate egg intake might suppress IHDi and IHDd on a global scale.

## 1. Introduction

Eggs are an important source of nutrients for humans, as they contain protein, fat, minerals, vitamins (except for vitamin C), and carotenoid pigments such as lutein [1]. Eggs are eaten worldwide in both developed and developing countries, as they are inexpensive and can be used in a variety of dishes [2].

The 2015 to 2020 Dietary Guidelines for Americans currently recommend the consumption of eggs as a dietary factor in a healthy diet [3]. However, eggs are the food with the highest cholesterol content, and high cholesterol is a known risk factor for atherosclerosis. Thus, it has been recommended for over 50 years that people limit their consumption of eggs, as these were thought to increase blood cholesterol levels, which is a risk factor for coronary artery disease.

However, it was subsequently found that dietary cholesterol had little effect on blood cholesterol concentrations, although the link between egg consumption and coronary heart disease has yet to be fully understood. Against this background, several epidemiological and clinical studies have been carried out worldwide to further evaluate the relationship between egg consumption and blood cholesterol concentrations, as well as between egg consumption and cardiovascular disease (CVD), including ischemic heart disease (IHD).

A meta-analysis found that there was no relationship between egg consumption and the risk of coronary heart disease [4]. Furthermore, a prospective cohort study in China found a negative association between egg consumption and cardiovascular disease [5]. In a recent cohort study of 30,835 people in China, a negative correlation between egg consumption and risk of death was also reported [6]. In contrast, a prospective cohort study of approximately 520,000 Americans and an updated meta-analysis in 2022 found that high egg consumption was associated with an increased risk of CVD [7,8,9]. Other studies have shown similar results, with these findings thought to be due to the increased cholesterol intake from egg consumption [8,10].

As mentioned above, current research results on the correlation between egg consumption and the risk of cardiovascular disease are inconsistent and inconclusive. In addition, most studies were conducted in Europe and North America, where there is a tendency towards over-nutrition. Only a few studies exist that have evaluated trends in other parts of the world, although 3 large international prospective studies of 177,000 people in 50 countries reported no significant associations between egg consumption and blood lipids, mortality, or CVD [11].

Epidemiologists and nutrition experts from around the world, such as the creators of the Global Burden of Disease (GBD) 2019 and the Global Dietary Database (GDD), have recently accelerated efforts to collect dietary data in order to improve global health, nutrition research, and policy [12,13]. These efforts have made it possible to understand the dietary habits of people worldwide. Furthermore, developments in analytical technology have made it possible to handle relatively sophisticated analyses using these data.

Our current study was designed to evaluate an international database and perform a longitudinal analysis of the association between egg consumption and ischemic heart disease incidence (IHDi) and mortality (IHDd) in 142 countries over 28 years from 1990 to 2018.

## 2. Methods

### 2.1. Incidence and Mortality of IHD

Age-standardized IHDi and IHDd rates per 1,000,000 persons in each country from 1990 to 2018 were obtained from the GBD 2019 database [12]. The GBD quantifies the impact of disease, injury, and risk factors in countries worldwide. These data are maintained by the Institute for Health Metrics and Evaluation (IHME) at the University of Washington, which created the database in conjunction with input from more than 5000 collaborators in 152 countries and territories.

### 2.2. Egg Intake

Egg intake in 1990, 1995, 2000, 2005, 2010, 2015, and 2018 was obtained from the GDD [13]. The GDD is an ongoing global dietary intake estimate produced by a team of epidemiologists and nutrition experts from around the world which is used to inform global health and nutrition research and policy. The GDD data provide estimates of individual food and nutrient intake according to country, year, and sex, for 185 countries.

### 2.3. Socioeconomic and Lifestyle Indicators

Several socioeconomic and lifestyle indicators were obtained from the GBD2019 [13], GDD, and World Bank databases [14] and used as covariates. Energy intake (kcal/day/capita), body mass index (BMI kg/m^2^), obesity rate (%), physical activity (1000 MET min/week), smoking rates (%), and years of education were obtained from the GBD2018. Dietary cholesterol intake (mg/day) was obtained from the GDD. Population (per million), aging rate (percentage of the population over 65 years), gross domestic product (GDP) per capita (USD/capita), and regional classification were obtained from the World Bank database. These are considered to be lifestyle and socioeconomic sector indicators that may be confounders of IHD.

### 2.4. Statistical Analysis

The longitudinal associations between egg intake and the incidence and mortality of IHD were examined using a linear mixed-effects model in one hundred forty-two countries with populations of one million or greater. In all cases, 28 years of data were available (from 1990 to 2018). The World Bank classifies the world into the following seven regions: (1) East Asia and the Pacific, (2) Europe and Central Asia, (3) Latin America and the Caribbean, (4) the Middle East and North Africa, (5) North America, (6) South Asia, and (7) Sub-Saharan Africa (Appendix A). The distribution of the global and regional variables in 2018 is presented as the mean and SD. The global variables are also presented with an interquartile range. The countries were divided according to this classification; then, the population-weighted average of egg intake in the region was calculated for each year. Subsequently, the data were plotted using the locally estimated scatterplot smoothing method (LOESS). To determine the association of egg intake with IHDi and IHDd, along with changes in this association according to the year, a linear mixed model analysis was performed; the IHDi and IHDd of each country over 28 years (from 1990 to 2018) were defined as dependent variables, while egg intake, year, and the interaction between egg intake and year were defined as independent variables.

In Model 1, year was used as a covariate, while year and GDP per capita were used as covariates in Model 2. In Model 3, in addition to year and GDP, aging rate, years of education, smoking rate, physical activity, mean BMI, energy supply, population, obesity rate, and dietary cholesterol intake were used as covariates. All independent variables were centralized using group means. After centralization, the year was divided by 100. The random effects of the mixed model were the intercept and the slope of the year for each country. In addition, the covariance matrix by year for each country specified a compound symmetric structure. The models were fitted by maximizing the log-likelihood. As this study aimed to evaluate repeated observations within countries, we used linear mixed models rather than regression analysis or repeated measures analysis, as these models can be applied without problems, even if the data contain missing values.

Our group has previously reported several studies using linear mixed models [15,16]. The Akaike information criterion (AIC) and the Bayesian information criterion (BIC) were used to estimate the relative quality of the models. R 4.0.5 was used for analysis [17]. A *p*-value < 0.05 was considered significant. The “lme” function of the “nlme” package was used for the generalized linear mixed-effects model [18].

## 3. Results

### 3.1. Characteristics of Variables in 2018

Table 1 shows the means, SDs, and percentiles for variables such as egg intake, IHDi and IHDd, and socioeconomic and lifestyle factors in 142 countries in 2018 according to region, as classified by the World Bank. The overall IHDi was 309.7 ± 188.4 per 100,000 persons, while IHDd was 146.5 ± 99.8 per 100,000 persons. The region with the lowest IHDi and IHDd rates was North America (187.4 ± 12.4, 76.9 ± 18.9 per 100,000 persons, respectively) while the Middle East and North Africa had the highest (604.2 ± 127.9, 216.3 ± 85.4 per 100,000 persons, respectively).

The mean and median egg intake were 21.5 g/day and 18.7 g/day, respectively. The 5th to 95th percentile for egg intake ranged from 3.5 to 45.1. The regions with the highest egg intake were the Middle East and North Africa (36.1 ± 22.1 g/day), while that with the lowest was Sub-Saharan Africa (6.9 ± 4.6 g/day).

### 3.2. Changes in Egg Intake and IHDi and IHDd

Figure 1 shows the global distribution of egg intake according to country in 2018. Egg intake data were obtained from the GDD. Countries for which no data on food intake were available for 2018 are shown in gray. These results show that egg intake occurs worldwide. The countries with the highest egg intake were Iraq (106.14 g/day), Mongolia (103.42 g/day), and Albania (87.37 g/day). The countries with the lowest egg intake were Angola (0.77 g/day), the Congo (0.99 g/day), and Rwanda (1.95 g/day), which are in South Africa (Appendix A).

Figure 2a shows the changes in egg intake and IHDi and IHDd that occurred globally and in the GBD regions from 1990 to 2018. The results demonstrate that the annual trends in egg intake varied from region to region. Global egg intake increased from 1990 to 2018. However, egg intake was lower in Sub-Saharan Africa and South Asia as compared to other regions, although the South Asian intake has increased in recent years. In particular, the Middle East and North Africa, East Asia, and the Pacific regions have shown dramatic increases similar to those observed in Latin America and the Caribbean. However, the global values for Europe and Central Asia and North America were flat.

Although the annual trends in global age-standardized IHDi and IHDd rates showed a global decline, differences were noted in the regional trends. Incidence and mortality were high in the Middle East and North Africa, while they were relatively low in Latin America and the Caribbean, East Asia, and the Pacific. In North America, they showed a dramatic decline over 25 years, and in the Middle East and North Africa, they also showed a gradual decline. In South Asia and Sub-Saharan Africa, they remained largely unchanged. There was a slight increase observed in East Asia and the Pacific.

### 3.3. Mixed-Effects Models of Egg Intake and IHD Incidence and Mortality Rate

Table 2 shows the results of the mixed-effects model analysis of the association between egg intake and IHDi. In Model 1, after controlling for the year as the main effect of egg intake as well as for standard error (β ± SE), a significant association was observed between egg intake and the incidence of IHD (−0.404 ± 0.119, *p* < 0.001). In Model 2, the main effects of egg intake when we controlled for the year and GDP showed that there was a significant association with the incidence of IHD (−0.358 ± 0.119, *p* < 0.01). In Model 3, after controlling for the year, GDP, aging rate, education, energy supply, smoking rate, BMI, physical activity, obesity rate, and dietary cholesterol intake, a significant correlation was observed between egg intake and the incidence of IHD (−0.253 ± 0.117, *p* < 0.05). The AIC and BIC were lowest and the log-likelihood highest in Model 3. Table 3 shows the results of the mixed-effects model analysis of the association between egg intake and IHDd. The results were similar to those for the incidence of IHD, with a significant negative correlation observed between egg intake and IHDd in Models 1 to 3. The main effect and standard error of the egg intake in Models 1 to 3 were −0.492 ± 0.136, *p* < 0.001; −0.429 ± 0.136, *p* < 0.01; and −0.358 ± 0.137, *p* < 0.05, respectively. The AIC and BIC were lowest and the log-likelihood highest in Model 3. In contrast, the interaction between egg intake and year demonstrated that there was no significant association (Appendix A).

## 4. Discussion

This study is the first long-term longitudinal study to demonstrate, using an international open database, that there is a significant negative association of egg intake with IHDi and IHDd. To the best of our knowledge, the long-term association between egg intake and ischemic heart disease has not been previously analyzed for such a large number of countries.

In our current study, upon analyzing egg intake over time according to regional group, we found that while eggs are eaten worldwide, there are regional differences with regard to their intake. However, since eggs are normally a locally produced and consumed agricultural product, it is possible that local cultural factors and environmental issues related to egg production, handling, and consumption influence egg intake [19]. Although egg intake is currently low in Asia and Africa, there has been a recent focus on egg-based nutrition as a means of addressing nutritional deficiencies in developing countries [19].Thus, egg intake in these areas may increase in the future.

Cardiovascular disease has long been a major contributor to the global burden of disease, with active intervention to address this issue proving to be a global challenge [20]. In the present study, it is clear from Figure 2 that the proportion of standardized IHDi and IHDd was high from 1990 to 2018.

When analyzed based on regional characteristics, IHDd and IHDi rates were recently found to have significantly decreased in North America. Decreases have also been observed in some regions while in others, such as the East Asia and Pacific regions, increases have been shown. Several studies have shown that switching to a healthier diet has an immediate positive impact on health, but many studies have shown that a healthy diet must be followed for weeks, months, or even years to reduce the risk of chronic diseases [21,22]. Therefore, changes in egg intake and in the incidence of cardiovascular disease and mortality must also be interpreted from a long-term perspective.

The results of the mixed-effect model analysis showed that egg intake was significantly associated with both IHDi and IHDd, and this did not change after controlling for either socioeconomic factors or lifestyle variables. These results are similar to those of several previous cohort studies that found egg consumption to be significantly associated with a lower risk of CVD [5,6]. Our analysis of the interaction between egg intake and year with regard to the relationship between egg intake and IHD also found that there was no association. Thus, these results suggest that the association between eggs and IHD is not influenced by the year.

The mechanism by which egg intake is negatively correlated with IHDi and IHDd may be related to the fact that egg intake does not increase blood cholesterol levels in most healthy people. Furthermore, the antioxidants found in eggs can contribute to the prevention of IHD.

Although the cholesterol concentrations of individual subjects were not included in our current analysis, many other observational and clinical trials have found no effect of egg intake on blood cholesterol concentrations in healthy subjects who have consumed as few as one or two eggs per day [23,24,25]. Although increased egg intake does increase cholesterol, some studies have suggested that the cholesterol in eggs is not readily absorbed, and thus, egg consumption does not increase blood cholesterol levels [26]. In addition, the component of the diet that has the greatest effect on blood cholesterol levels is not dietary cholesterol, but rather has been shown to be dietary saturated fatty acids [27]. Generally, foods high in cholesterol are high in saturated fatty acids, yet eggs are low in saturated fatty acids [28]. Therefore, the risk of elevated blood cholesterol may be lower for egg intake compared to other animal products.

It has also been shown that oxidative stress plays a role in the development of cardiovascular diseases [29], and that the carotenoids and vitamin E present in eggs have a strong antioxidant effect due to their chemical structure, which protects the body from oxidative stress. Intervention studies conducted in healthy adults have shown that daily egg consumption increases antioxidant indices [24], increases blood carotenoid levels, and inhibits the oxidation of LDL cholesterol, the latter of which is a cause of atherosclerosis [30]. It has also been reported that the absorption of carotenoids and vitamin E contained in fruits and vegetables was enhanced by eating eggs [31,32]. This suggests that there may be an interaction between eggs and the ingredients eaten with them.

The results of our current study show that the higher the egg intake, the lower the age-standardized incidence of ischemic heart disease and the lower the mortality rate. However, these results should be interpreted with caution. Additionally, it should be noted that it is not our intention to recommend a high intake of eggs, which contain relatively high levels of cholesterol, for the prevention of IHD, as high levels of cholesterol in the blood are clearly a cause of dyslipidemia [33]. The conclusions of this study are opposed to the findings of many prospective cohort studies that have reported a positive association between egg consumption and the risk of cardiovascular disease or total mortality [8,9]. The reason for the different results from such cohort studies may be related to the fact that the data used in this study included an average egg intake of 21.5 ± 17.1 g/day, with a maximum of 106.14 g/day. One egg weighs about 50–60 g, so the maximum is within the range of one or two eggs. This is lower than the level of intake in Europe and the USA, where many analyses of the association between eggs and ischemic heart disease have been conducted. The findings of the present study, those of many other researchers who recommend egg intake [30,34,35,36], and the American Heart Association recommend one or two eggs per day as part of a healthy diet [37]. Thus, overall, it seems appropriate to discuss the acceptability of egg intake in the range of one or two eggs per day.

In regional terms, there may be potential health benefits from increased egg consumption in the Sub-Saharan and South Asian regions, where intake is currently low. The EAT-Lancet Committee on Healthy Eating has stated that higher egg consumption may be beneficial to health in low-income groups with poor diets [38]. Moreover, it has been shown that there has been a recent increase in egg consumption in regions with high IHDi and IHDd, such as the Middle East and North Africa. Three large international prospective studies in 50 countries on six continents reported that egg consumption did not increase blood lipids or cardiovascular events [11], and further studies are expected to determine whether egg consumption will have a beneficial effect on future IHDi and IHDd in these regions.

There are several limitations of our current study. First, there were methodological limitations in that the design of our study was ecological in nature. The collected data were obtained from an international database of countries, and did not consider individual differences in age, sex, or lifestyle. Therefore, these results do not allow us to show causality at an individual level. Secondly, our study did not take into account the influence of other nutritional factors that could potentially contribute to ischemic heart disease. In addition, we did not take into account any of the foods that were consumed with the eggs or the ways in which eggs were prepared, as this can vary from country to country. Furthermore, there are two major factors that contribute to IHD risk reduction: lifestyle, such as diet, and advancements in medical technology. It is important to combine both medical and lifestyle interventions, such as diet, to reduce IHD risk in any region of the world. However, this study did not consider the quality of medical technology itself or improvements in technology. In contrast, the strength of our study is that we have included countries and regions that have not been analyzed before, whereas most of the previous research in this field was conducted in Europe, the United States, Scandinavia, Japan, and China.

## 5. Conclusions

In conclusion, the results of our comparison using an international database showed that egg consumption was associated with lower IHDi and IHDd. Considering their combination of nutritional benefits, low cost, and culinary versatility, eggs have the potential to contribute to healthy diets in many countries worldwide.

## Figures and Tables

**Figure 1 ijerph-20-04138-f001:**
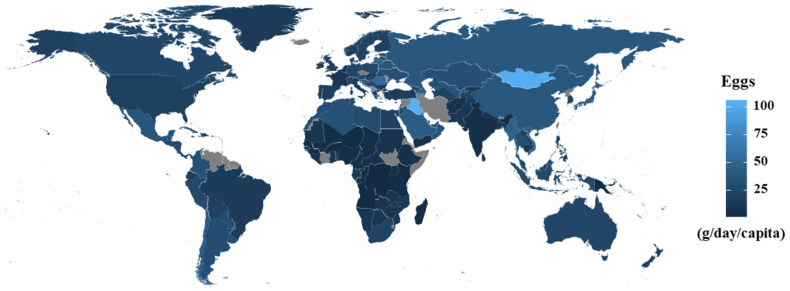
Global distribution of egg intake by country. Egg intake data were obtained from the Global Dietary Database. Countries shown in gray indicate areas where no food intake data were obtained during 2018.

**Figure 2 ijerph-20-04138-f002:**
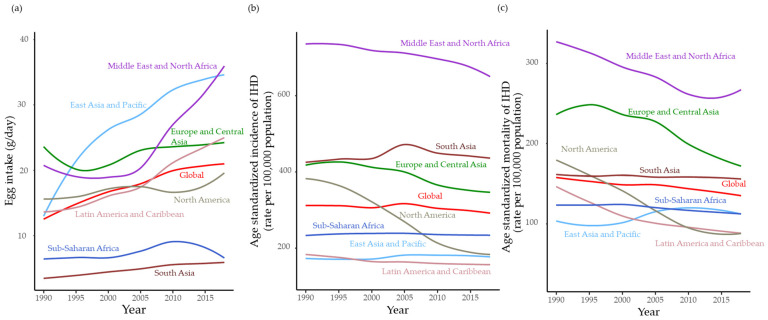
Global and regional changes in egg intake (**a**), ischemic heart disease (IHD) incidence (**b**), and mortality (**c**), from 1990 to 2018.

**Table 1 ijerph-20-04138-t001:** Mean value, standard deviation, and percentiles of IHD incidence and mortality, socioeconomic variables, lifestyle variables, and egg intake in 142 countries.

	Overall	World Bank Regions							
		East Asia and Pacific	Europe and Central Asia	Latin America and Caribbean	Middle East and North Africa	North America	South Asia	Sub-Saharan Africa	*p*-Value
	N = 142	N = 17	N = 42	N = 21	N = 16	N = 2	N = 6	N = 38	
Egg intake (g/day)	21.5 ± 17.1 (3.5–45.1)	30.3 ± 22.2	25.6 ± 14.5	25.2 ± 8.9	36.1 ± 22.1	21.8 ± 3.7	8.7 ± 3.3	6.9 ± 4.6	<0.001
Population (million)	51 ± 167 (1.9–160.5)	133 ± 331	21 ± 30	29 ± 50	22 ± 25	182 ± 205	302 ± 521	25 ± 36	<0.001
Aging rate (%)	9.0 ± 6.6 (2.4–20.4)	9.3 ± 6.3	16.1 ± 5.5	8.5 ± 2.9	4.6 ± 2.9	16.5 ± 1.0	5.7 ± 2.6	3.2 ± 1.5	<0.001
GDP (1000 USD/capita)	14.43 ± 19.38 (0.6–53.7)	17.12 ± 21.70	25.90 ± 23.23	8.28 ± 5.03	18.27 ± 19.70	54.58 ± 11.69	1.80 ± 1.23	2.23 ± 2.82	<0.001
Education (years)	8.8 ± 3.2 (3.4–13.2)	8.8 ± 3.1	12.0 ± 1.2	8.6 ± 1.8	9.0 ± 2.1	13.0 ± 0.2	5.7 ± 2.0	5.7 ± 2.3	<0.001
Smoking rate (%)	18.4 ± 7.8 (7.5–31.2)	23.3 ± 5.5	25.1 ± 5.6	14.6 ± 7.2	17.1 ± 5.4	16.8 ± 0.2	16.1 ± 5.0	12.0 ± 5.2	<0.001
Physical activity (1000 MET min/week)	5.4 ± 1.8 (2.9–8.7)	6.6 ± 2.4	5.2 ± 1.5	4.7 ± 1.2	3.4 ± 0.9	5.1 ± 0.4	6.3 ± 1.9	6.2 ± 1.6	<0.001
Body mass index (kg/m^2^)	25.6 ± 2.2 (21.8–28.7)	24.0 ± 2.0	26.6 ± 0.9	26.7 ± 1.3	28.1 ± 1.8	28.8 ± 1.1	23.1 ± 0.8	23.7 ± 1.5	<0.001
Energy supply (1000 kcal/capita/day)	2.66 ± 0.43 (2.0–3.4)	2.60 ± 0.33	2.87 ± 0.31	2.55 ± 0.35	3.14 ± 0.40	3.34 ± 0.18	2.53 ± 0.23	2.30 ± 0.29	<0.001
Obesity rate (%)	15.3 ± 8.5 (8.5–29.9)	9.4 ± 7.3	17.9 ± 4.0	17.6 ± 5.6	27.8 ± 9.4	27.4 ± 6.4	6.3 ± 1.8	9.3 ± 5.3	<0.001
Cholesterol intake (mg/day)	233.6 ± 88.5 (92.7–375.3)	266.7 ± 61.7	291.6 ± 61.7	274.8 ± 57.2	277.5 ± 51.7	264.7 ± 18.5	88.2 ± 35.4	134.8 ± 36.2	<0.001
IHD incidence (/100,000/year)	309.7 ± 188.4 (108.4–651.8)	203.1 ± 128.0	351.8 ± 204.5	207.5 ± 113.0	604.2 ± 127.9	187.4 ± 12.4	417.3 ± 169.2	232.9 ± 70.5	<0.001
IHD mortality (/100,000/year)	146.5 ± 99.8 (49.0–337.1)	115.7 ± 68.1	179.4 ± 147.3	100.1 ± 41.1	216.3 ± 85.4	76.9 ± 18.9	168.3 ± 81.6	120.6 ± 33.0	<0.001

The figures in this table are the results for 2018. Data are presented as means ± SD (interquartile range). GDP: gross domestic product; IHD: ischemic heart disease.

**Table 2 ijerph-20-04138-t002:** Main effects of egg intake and covariables on the incidence of IHD, determined using linear mixed-effects models.

	Model 1	Model 2	Model 3
IHDi	β SE	β SE	β SE
(Intercept)	327.590 (15.916) ***	327.704 (15.893) ***	322.598 (17.803) ***
Eggs	−0.404 (0.119) ***	−0.358 (0.119) **	−0.253 (0.117) *
Year	−1.080 (0.238) ***	−0.798 (0.236) ***	−0.803 (0.476)
GDP		−0.868 (0.223) ***	−0.381 (0.220)
Aging rate			1.686 (1.121)
Education			9.420 (3.620) **
Energy supply			−77.355 (13.609) ***
Smoking			1.305 (0.557) *
BMI			19.172 (6.492) **
Physical activity			−31.314 (8.576) ***
Obesity			−6.623 (1.298) ***
Cholesterol intake			0.029 (0.028)
AIC	9515.327	9504.598	9392.431
BIC	9554.304	9548.438	9475.098

***: *p* < 0.001. **: *p* < 0.01. *: *p* < 0.05. IHDi: ischemic heart disease incidence; GDP: gross domestic product; BMI: body mass index; AIC: Akaike information criterion; BIC: Bayesian information criterion; SE: standard error.

**Table 3 ijerph-20-04138-t003:** Main effects of egg intake and covariables on the mortality of IHD, determined using linear mixed-effects models.

	Model 1	Model 2	Model 3
IHDd	β SE	β SE	β SE
(Intercept)	169.579 (8.502) ***	169.694 (8.359) ***	163.269 (11.551) ***
Eggs	−0.492 (0.136) ***	−0.429 (0.136) **	−0.359 (0.137) **
Year	−1.690 (0.259) ***	−1.329 (0.255) ***	−1.189 (0.413) **
GDP		−1.106 (0.239) ***	−0.736 (0.251) **
Aging rate			1.354 (1.186)
Education			8.129 (3.023) **
Energy supply			−73.211 (14.602) ***
Smoking			1.064 (0.608)
BMI			17.016 (6.890) *
Physical activity			−9.450 (5.419)
Obesity			−5.797 (1.425) ***
Cholesterol intake			0.033 (0.032)
AIC	9573.452	9557.163	9477.264
BIC	9612.429	9601.003	9559.931

***: *p* < 0.001. **: *p* < 0.01. *: *p* < 0.05. IHDd: ischemic heart disease mortality; GDP: gross domestic product; BMI: body mass index; AIC: Akaike information criterion; BIC: Bayesian information criterion; SE: standard error.

## Data Availability

The data associated with this paper are not publicly available but are available from the corresponding author upon reasonable request.

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
