# Peer review of "The Global Association between Egg Intake and the Incidence and Mortality of Ischemic Heart Disease—An Ecological Study"

_ijerph, 2023, doi:10.3390/ijerph20054138_

Round 1

Reviewer 1 Report

Nice job overall, please take into account the following:

1. Please add a simple summary section before the abstract

2. In statistical analysis, can you show the model used to analyze the data?

3. Do you think that quality of medical services has anything to do with the reduction of mortalities due to IHD in North America?

Reviewer 2 Report

Congratulations on the work, I think it is a very powerful study although the ecological design limits it in reaching conclusions. Have you considered analyzing the differences in the association between egg intake and cardiovascular events according to different geographical areas?

I have some comments-doubts:

In line 143 there is a sentence that is subsequently repeated.

Table 1 has some regions highlighted in bold. Check if this is correct. If yes, why?

In the discussion you could add how changes in lifestyle or dietary patterns (egg intake) may take time to be reflected in the incidence of cardiovascular events. This should be taken into account when interpreting Figure 2.

Reviewer 3 Report

Dear authors, 

The topic is very interesting and usuful, since it offers some clarifications for many readers, scientists or not. But, in order to be published, some improvements are required. The introduction section needs an update with previous studies from other countries and the discussion section must consist comments regarding previous researches. 
